# Accurate Brain Age Prediction from MRI: Evaluating Kolmogorov-Arnold and Convolutional Networks

**Alessandro Giupponi**[*1]  ⓘD         ALESSANDRO.GIUPPONI@PHD.UNIPD.IT

**Davide De Crescenzo**[*1]          DAVIDE.DECRESCENZO@STUDENTI.UNIPD.IT

**Marco Pinamonti**[1]           MARCO.PINAMONTI.1@PHD.UNIPD.IT

**Manuela Moretto**[1,2]          MANUELA.MORETTO@UNIPD.IT

**Alessandra Bertoldo**[1,3]        ALESSANDRA.BERTOLDO@UNIPD.IT

**Mattia Veronese**[1,2]          MATTIA.VERONESE@UNIPD.IT

**Marco Castellaro**[1]  ⓘD         MARCO.CASTELLARO@UNIPD.IT

[1] *Department of Information Engineering, University of Padova, Padova, Italy*

[2] *Neuroimaging Department, IoPPN, King's College London, London, UK*

[3] *Padova Neuroscience Center, University of Padova, Padova, Italy*

**Editors:** Accepted for publication at MIDL 2025

## Abstract

Brain age prediction using T1-weighted MRI has become a key biomarker for assessing neurological health, with application in studying neurodegeneration (Soumya Kumari and Sundarrajan, 2024; Mishra et al., 2023; Lea et al., 2021) and brain development (Tanveer et al., 2023). While convolutional neural networks (CNNs) remain a standard approach, recent advances suggest that Kolmogorov-Arnold Networks (KANs) may offer superior performance in image-based task (Bodner et al., 2025; Li et al., 2024). In this study, we present the first use of KANs for brain age prediction from 3D MRI scans, comparing their performance against traditional CNNs. Experimental results show that KAN-based models reduce estimation errors, highlighting their potential for improving brain age assessment.

**Keywords:** Brain aging, Convolutional Neural Networks, Kolmogorov-Arnold Networks, Neurological biomarker

## 1. Introduction

Brain age prediction serves as a valuable biomarker, offering insights into neurodegenerative disorders, cognitive decline, and the effects of lifestyle on aging (Natalia et al., 2024; Franke and Gaser, 2019; Dias et al., 2025). Deep learning models, particularly CNNs, have been widely applied to this task due to their capacity to extract meaningful features from MRI scans (Peng et al., 2021; Dartora et al., 2024; Dinsdale et al., 2021). However, recent advances in neural architectures, such as Kolmogorov-Arnold Networks (Liu et al., 2025), provide new opportunities to enhance prediction accuracy (Patel et al., 2024).

KANs utilize the Kolmogorov-Arnold representation theorem to approximate complex functions (Schmidt-Hieber, 2020) more efficiently than conventional neural networks (SS et al., 2024; Yeo et al., 2025). They have shown promising results in classification, segmentation, and image generation tasks. This study investigates the application of convolutional KANs and hybrid CNN-KAN models for brain age prediction, comparing their performance to traditional CNNs.

---

[*] Contributed equally

## 2. Materials and Methods

**Dataset.** T1-weighted MRI scans were sourced from three public datasets: the Human Connectome Project (Bookheimer et al., 2019), the Nathan Kline Institute - Rockland Sample (Nooner et al., 2012), and the Cambridge Centre for Aging and Neuroscience (Taylor et al., 2017), totaling 2,129 participants (878 males, 1,250 females), aged 18–100. To ensure consistent input dimension (193 x 229 x 193) and spatial alignment across datasets, all images were linearly coregistered to the MNI152 2009c standard space; no further harmonization was performed. To enhance model robustness, data augmentation (DA) techniques, including rotation ($\pm\,40°$) and translation ($\pm\,10$ pixels), were applied (Connor and M., 2019). The dataset was randomly split into training (64%), validation (16%), and test (20%) subsets, preserving age and sex distributions. A subset of experiment used cross-validation, where training/validation splits were randomly pooled preserving age and sex distributions across folds. Mann-Whitney U tests confirmed no statistical age/sex differences between training and test sets (p = 0.901) nor between training and validation sets across cross-validation folds (lowest p = 0.840).

**Models Architecture.** Three models were tested: a baseline 3D CNN (Cole et al. (2017)) (**CNN**), a 3D convolutional KAN with a linear KAN output (**KAN**), and a hybrid 3D CNN with a final fully connected linear KAN layer (**CNN + KAN-Lin**). All models used 3x3x3 convolutional kernels; both stride 1 and 2 for the first convolutional layer were tested for CNNs, while KAN used only stride 2 due to high memory requirements.

**Training and Evaluation.** Models were trained to minimize Mean Squared Error (MSE) between true and predicted brain age using Adam (learning rate $10^{-4}$) over 1,000 epochs with validation every 50. Five-fold cross-validation was used only for stride-2 models. Final models (lowest validation loss) were evaluated on the test set using Mean Absolute Error (MAE) and Pearson Correlation Coefficient (PCC). For cross-validation, performance metrics were obtained via median ensembling of the best models across folds.

## 3. Results and Discussion

As shown in Table 1, the stride-2 KAN model outperformed the CNN, reducing error by 12.40%. The CNN + KAN-Lin hybrid, with DA, improved accuracy by 11.72% and offered the best trade-off between performance and computational load. Moreover, data augmentation consistently enhanced generalizability across all models, improving test performance.

| Stride | Method | Without DA | | With DA | |
|---|---|---|---|---|---|
| | | MAE | r | MAE | r |
| 2 | CNN | 5.982 | 0.908 | 4.588 | 0.947 |
| | KAN | 5.240 | 0.930 | 4.561 | 0.946 |
| | CNN + KAN-Lin | 5.286 | 0.932 | **4.051** | **0.959** |
| 1 | CNN | 4.929 | 0.944 | 4.158 | 0.958 |
| | CNN + KAN-Lin | 4.994 | 0.943 | **3.918** | **0.962** |

Table 1: MAE and PCC obtained for the different models with and without the use of data augmentation.

Due to memory constraints, stride-1 evaluations excluded the KAN model. Nonetheless, the hybrid model still outperformed CNN by 5.77%, though the margin was smaller—likely because higher resolution input enabled the CNN layer to extract finer features, reducing the added value of the KAN layer.

Both models showed age-related bias (Figure 1), overestimating younger and underestimating older subjects. The hybrid model yielded a smoother Predicted Age Difference (PAD) curve and improved accuracy in middle-aged individuals, but residual biases persisted at the extremes ($\leq 30$ and $\geq 70$ years), suggesting a need for improved age-related features or bias mitigation strategies.

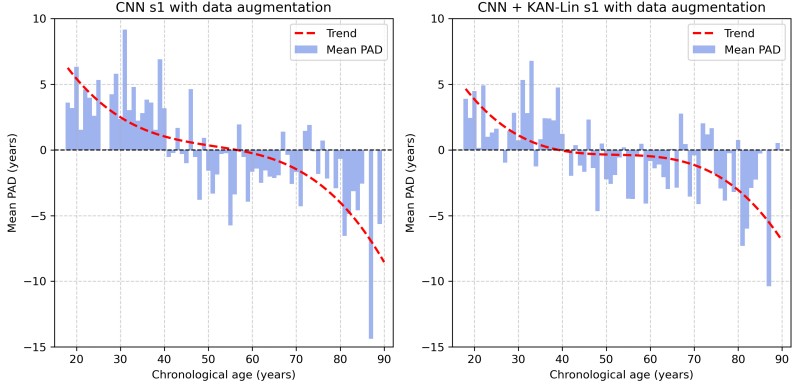

Figure 1: Mean PAD across chronological age bins (5-year intervals) for two models (left: CNN stride-1, right: CNN + KAN-Lin stride-1) trained with data augmentation. The red dashed line represents a third-order polynomial fit of PAD.

## 4. Conclusion

This study demonstrates the potential of Kolmogorov-Arnold Networks for brain age prediction from T1-weighted MRI. While KAN achieved the highest accuracy, the CNN + KAN-Lin hybrid offered the best balance of performance and efficiency, showing strong generalizability with data augmentation. Despite persistent age-related bias—overestimating younger and underestimating older subjects—the hybrid yielded smoother PAD curves and better accuracy in middle-aged groups by combining CNN spatial feature extraction with KAN's functional modeling. Future work should address extreme-age bias, possibly via targeted regularization or debiasing. These findings support integrating KANs into neuroimaging pipelines and opens the door to exploring their broader use in medical imaging.

## Acknowledgments

This project was supported by the Ministry of University and Research within the Complementary National Plan PNC-I.1 "Research initiatives for innovative technologies and pathways in the health and welfare sector, D.D. 931 of 06/06/2022, PNC0000002 DARE - Digital Lifelong Prevention CUP: B53C22006440001.

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
