# OpenReview forum: "Accurate Brain Age Prediction from MRI: Evaluating Kolmogorov-Arnold and Convolutional Networks"
_MIDL.io/2025/Short_Papers — MIDL 2025 - Short Papers_

### Official Review · Reviewer_wAJa · 2025-04-25

**Rating:** 4
**Confidence:** 5

**Summary:**

This study explores the use of Kolmogorov-Arnold Networks (KANs) for predicting brain age from T1-weighted MRI scans, comparing their performance to traditional Convolutional Neural Networks (CNNs). Results showed that KANs and a hybrid CNN + KAN-Lin model outperformed standard CNNs, especially when combined with data augmentation. The hybrid model achieved the promising balance between prediction accuracy and computational efficiency.

**Strengths:**

A key strength of the study is the introduction of KANs to brain imaging tasks, demonstrating improved accuracy over CNNs. The use of multiple large datasets, extensive data augmentation, and five-fold cross-validation strengthened the model's generalizability and robustness across different age groups.

**Weaknesses:**

Despite improved performance, the models still showed age-related prediction bias, consistently overestimating younger ages and underestimating older ones. Additionally, the KAN model's high computational demands limited its evaluation at higher input resolutions.

---

### Decision · Program_Chairs · 2025-05-01

Accept